# Vitamin D Supplementation Does Not Influence SARS-CoV-2 Vaccine Efficacy or Immunogenicity: Sub-Studies Nested within the CORONAVIT Randomised Controlled Trial

**DOI:** 10.3390/nu14183821

**Published:** 2022-09-16

**Authors:** David A. Jolliffe, Giulia Vivaldi, Emma S. Chambers, Weigang Cai, Wenhao Li, Sian E. Faustini, Joseph M. Gibbons, Corinna Pade, Anna K. Coussens, Alex G. Richter, Áine McKnight, Adrian R. Martineau

**Affiliations:** 1Wolfson Institute of Population Health, Barts and the London School of Medicine and Dentistry, Queen Mary University of London, London E1 2AB, UK; 2Blizard Institute, Barts and the London School of Medicine and Dentistry, Queen Mary University of London, London E1 2AT, UK; 3Institute of Immunology and Immunotherapy, College of Medical and Dental Sciences, University of Birmingham, Birmingham B15 2TT, UK; 4Infectious Diseases and Immune Defence Division, Walter and Eliza Hall Institute of Medical Research, Parkville 3052, Australia; 5Institute of Infectious Disease and Molecular Medicine, University of Cape Town, Cape Town 7925, South Africa; 6Asthma UK Centre for Applied Research, Queen Mary University of London, London E1 2AB, UK

**Keywords:** vitamin D, ChAdOx1 nCoV-19 Oxford–AstraZeneca, BNT162b2 Pfizer, breakthrough SARS-CoV-2 infection, randomised controlled trial, antibody, interferon gamma

## Abstract

Vitamin D deficiency has been reported to associate with the impaired development of antigen-specific responses following vaccination. We aimed to determine whether vitamin D supplements might boost the immunogenicity and efficacy of SARS-CoV-2 vaccination by conducting three sub-studies nested within the CORONAVIT randomised controlled trial, which investigated the effects of offering vitamin D supplements at a dose of 800 IU/day or 3200 IU/day vs. no offer on risk of acute respiratory infections in UK adults with circulating 25-hydroxyvitamin D concentrations <75 nmol/L. Sub-study 1 (*n* = 2808) investigated the effects of vitamin D supplementation on the risk of breakthrough SARS-CoV-2 infection following two doses of SARS-CoV-2 vaccine. Sub-study 2 (*n* = 1853) investigated the effects of vitamin D supplementation on titres of combined IgG, IgA and IgM (IgGAM) anti-Spike antibodies in eluates of dried blood spots collected after SARS-CoV-2 vaccination. Sub-study 3 (*n* = 100) investigated the effects of vitamin D supplementation on neutralising antibody and cellular responses in venous blood samples collected after SARS-CoV-2 vaccination. In total, 1945/2808 (69.3%) sub-study 1 participants received two doses of ChAdOx1 nCoV-19 (Oxford–AstraZeneca); the remainder received two doses of BNT162b2 (Pfizer). Mean follow-up 25(OH)D concentrations were significantly elevated in the 800 IU/day vs. no-offer group (82.5 vs. 53.6 nmol/L; mean difference 28.8 nmol/L, 95% CI 22.8–34.8) and in the 3200 IU/day vs. no offer group (105.4 vs. 53.6 nmol/L; mean difference 51.7 nmol/L, 45.1–58.4). Vitamin D supplementation did not influence the risk of breakthrough SARS-CoV-2 infection in vaccinated participants (800 IU/day vs. no offer: adjusted hazard ratio 1.28, 95% CI 0.89 to 1.84; 3200 IU/day vs. no offer: 1.17, 0.81 to 1.70). Neither did it influence IgGAM anti-Spike titres, neutralising antibody titres or IFN-γ concentrations in the supernatants of S peptide-stimulated whole blood. In conclusion, vitamin D replacement at a dose of 800 or 3200 IU/day effectively elevated 25(OH)D concentrations, but it did not influence the protective efficacy or immunogenicity of SARS-CoV-2 vaccination when given to adults who had a sub-optimal vitamin D status at baseline.

## 1. Introduction

Vaccination against SARS-CoV-2 represents the mainstay of COVID-19 control. However, vaccine efficacy and effectiveness wane significantly within 6 months, particularly among older adults [1]. Identification of immunomodulatory adjuvants with the potential to augment SARS-CoV-2 vaccine immunogenicity is therefore a research priority [2]. Sub-optimal responses to vaccination against other pathogens in older adults are causally associated with increased systemic inflammation, termed ‘inflammaging’ [3]. Increased production of inflammatory cytokines by monocytes and macrophages is a key driver of this process [4], and the pharmacological inhibition of these pathways by blocking p38 mitogen-activated protein (MAP) kinase or the mammalian target of the rapamycin (mTOR) pathway has been shown to augment antigen-specific immunity [5,6,7].

Vitamin D is best known for its effects on calcium homeostasis, but it is also recognised to play a key role in the regulation of human immune function [8]. The active vitamin D metabolite 1,25-dihydroxyvitamin D (1,25[OH]_2_D) has been shown to inhibit the production of pro-inflammatory cytokines by monocytes and macrophages by targeting MAP kinase phosphatase 1 [9], to regulate the mTOR pathway [10] and to support classical T cell receptor signalling and T cell activation by inducing phospholipase C-gamma 1 in naïve T cells [11]. Sub-optimal vitamin D status, as indicated by low circulating concentrations of 25-hydroxyvitamin D (25[OH]D, the major circulating vitamin D metabolite) is common among older adults, and this associates with increased systemic inflammation [12,13]. An experimental study has demonstrated that vitamin D supplementation significantly increased the response to the cutaneous varicella zoster virus (VZV) antigen challenge in older adults with circulating 25(OH)D concentrations less than 75 nmol/L [14]. This enhancement was associated with a reduction in early inflammatory monocyte infiltration with a concomitant enhancement of T cell recruitment to the site of the antigen challenge.

Taken together, these findings provide a rationale for investigating whether vitamin D replacement might enhance the immunogenicity and effectiveness of SARS-CoV-2 vaccination in adults with sub-optimal vitamin D status [15,16]. Several observational studies have investigated associations between vitamin D status and SARS-CoV-2 vaccine immunogenicity, but these have yielded conflicting results: some report higher post-vaccination titres of anti-Spike antibodies in individuals using vitamin D supplements or having higher circulating 25(OH)D concentrations [17,18], but others have yielded null findings [19,20]. An opportunity to investigate this question using an interventional study design arose when we conducted a phase 3 randomised controlled trial of vitamin D supplements for prevention of acute respiratory infection in UK adults (CORONAVIT) [21]. The intervention period for this study coincided with the rollout of SARS-CoV-2 vaccination over Winter–Spring 2020–21; a period when sub-optimal vitamin D status was highly prevalent in the UK [22]. We therefore nested three sub-studies within the trial to investigate the effects of vitamin D replacement on SARS-CoV-2 vaccine efficacy, post-vaccination titres of anti-Spike antibodies in dried blood spot eluates and post-vaccination neutralising antibody titres and antigen-specific cellular responses to SARS-CoV-2 in venous blood.

## 2. Materials and Methods

### 2.1. Study Design

We conducted three sub-studies nested within the CORONAVIT randomised controlled trial [21]. Sub-study 1 (vaccine efficacy analysis) investigated the influence of vitamin D supplementation on the risk of breakthrough SARS-CoV-2 infection in immunocompetent trial participants who received two doses of a SARS-CoV-2 vaccine during follow-up. Sub-study 2 (dried blood spot analysis) investigated the effects of vitamin D supplements on combined IgG, IgA and IgM (IgGAM) antibody responses to the Spike (S) protein of SARS-CoV-2 measured in dried blood spot eluates. Sub-study 3 (venous blood analysis) investigated the effects of vitamin D supplements on neutralising antibody and cellular responses.

Full details relating to the design and conduct of the CORONAVIT trial and post-vaccination serology studies have been reported elsewhere [17,21]. Briefly, 6200 UK residents aged 16 years or older and participating in the COVIDENCE UK study [23] were individually randomised to receive an offer of a postal vitamin D test, followed by higher-dose (3200 IU/day; *n* = 1550) or lower-dose (800 IU/day; *n* = 1550) vitamin D supplementation if their blood 25(OH)D concentration was found to be less than 75 nmol/L, or to receive no offer of vitamin D testing or supplementation (*n* = 3100), with a 1:1:2 allocation ratio. Treatment allocation was not concealed, and randomisation was not stratified. All participants who received at least two doses of a SARS-CoV-2 vaccine were invited to provide a postal dried blood spot sample for the determination of combined IgGAM antibody responses to the S protein of SARS-CoV-2, as described below. A subset of 101 trial participants also provided a venous blood sample for the determination of neutralising antibody and cellular immune responses to SARS-CoV-2. The trial was sponsored by Queen Mary University of London, approved by the Queens Square Research Ethics Committee, London, UK (ref 20/HRA/5095) and registered with ClinicalTrials.gov (NCT04579640) on 8 October 2020, before enrolment of the first participant on 28 October 2020.

### 2.2. Participants

Eligibility criteria for the three sub-studies were as follows. For sub-study 1 (vaccine efficacy analysis), inclusion criteria were participation in the CORONAVIT trial and receipt of two doses of a SARS-CoV-2 vaccine, with the first dose given between 16 January 2021 (i.e., at least 1 month after the start of the trial) and 16 June 2021 (i.e., the end of the trial intervention period). Exclusion criteria for sub-study 1 were self-report of taking study vitamin D capsules less than half the time during trial follow-up (intervention arms) or self-report of any intake of supplemental vitamin D during follow-up (no-offer arm); known immunodeficiency disorder; and use of systemic immunosuppressants. For sub-study 2 (dried blood spot analysis), inclusion criteria were eligibility for sub-study 1, plus consent to participate in the post-vaccination serology sub-study and availability of an anti-S titre result from a dried blood spot sample provided at least 2 weeks after administration of the second dose of a SARS-CoV-2 vaccine and before administration of a booster dose. For sub-study 3 (venous blood analysis), inclusion criteria were eligibility for sub-study 1, plus residence within a 100-mile radius of the Blizard Institute (East London), consent to participate in the post-vaccination venous blood sub-study and availability of valid neutralising antibody or cellular response data.

### 2.3. Randomisation

CORONAVIT trial participants were individually randomised by the trial statistician to either higher-dose offer, lower-dose offer or no offer using a computer program (Stata version 14.2; College Station, TX, USA), as previously described [21].

### 2.4. Intervention

Consenting participants randomised to either intervention arm of the trial were posted a blood spot testing kit for the determination of 25(OH)D concentrations in capillary blood, as previously described [24]. Those found to have a 25(OH)D concentration below 75 nmol/L were then posted a 6-month supply of capsules containing either 800 IU or 3200 IU vitamin D_3_, according to their allocation. Participants were supplied with D-Pearls capsules of either strength, manufactured by Pharma Nord Ltd. (Vejle, Denmark), unless they expressed a preference for a vegetarian or vegan supplement, in which case they were supplied with Pro D_3_ vegan capsules manufactured by Synergy Biologics Ltd. (Walsall, UK). Participants with 25(OH)D concentrations of 75 nmol/L or more at initial testing were offered a second postal vitamin D test 2 months after the first test: those whose second 25(OH)D concentration was found to be less than 75 nmol/L were offered a postal supply of supplements as above. Participants receiving study supplements were instructed to take one capsule per day until their supply was exhausted. The administration of study supplements was not supervised.

### 2.5. Follow-Up Assessments

Follow-up for breakthrough SARS-CoV-2 infection (primary outcome, sub-study 1) was from 2 weeks after the second dose of SARS-CoV-2 vaccine up to 6 months thereafter or the date of the breakthrough SARS-CoV-2 infection, whichever was earlier. Vaccination details and breakthrough SARS-CoV-2 infections confirmed by RT-PCR or antigen testing were captured via online questionnaires sent to all participants at monthly intervals and complemented by electronic linkage to routinely collected medical record data, as previously described [21]. Every monthly questionnaire contained the following advice to encourage participants with COVID-19 symptoms to engage with testing services: “If you currently have symptoms of coronavirus (a high temperature, a new, continuous cough or loss of or altered sense of smell or taste), call NHS111 or visit https://www.nhs.uk/conditions/coronavirus-covid-19/ (accessed on 17 August 2022) for more information.” This wording was identical for questionnaires sent to participants randomised to intervention or no-offer groups. In addition to monthly questionnaires, an online adherence questionnaire was sent to all participants randomised to either offer on 31 March 2021. This questionnaire captured information regarding the frequency of study supplement use. End-trial postal vitamin D testing was offered to a randomly selected subset of 1600 participants who received study supplements (800 participants from each intervention group) and 400 who were randomised to no offer. Participants randomised to no offer who were found to have end-trial total 25(OH)D concentrations below 50 nmol/L were posted a 60-day supply of capsules, each containing 2500 IU vitamin D_3_ (Cytoplan Ltd. Worcestershire, England).

### 2.6. Laboratory Assays

#### 2.6.1. 25(OH)D Testing

In total, 25(OH)D assays were performed by Black Country Pathology Services, located at Sandwell General Hospital, West Bromwich, UK; this laboratory participates in the UK NEQAS for Vitamin D and the Vitamin D External Quality Assessment Scheme (DEQAS) for serum 25(OH)D. Concentrations of 25(OH)D_3_ and 25(OH)D_2_ were determined in dried blood spot eluates using liquid chromatography tandem mass spectrometry (Acquity UPLC-TQS or TQS-Micro Mass Spectrometers, Waters Corp., Milford, MA, USA) after derivatisation and liquid–liquid extraction as previously described [24] and summed to give total 25(OH)D concentrations. Very good overall agreement between blood spot and plasma 25(OH)D concentrations in paired capillary and venous samples using this blood spot method has been observed [24], demonstrating a minimal overall bias of −0.2% with a bias range of −16.9% to 26.7%. Total 25(OH)D concentrations lower than 75 nmol/L were defined as sub-optimal: this threshold is widely considered to discriminate between those with lower vs. higher vitamin D status [25,26,27]. The between-day coefficients of variation were 11.1% at 16.9 nmol/L, 8.2% at 45.5 nmol/L, 6.9% at 131.7 nmol/L and 7.0% at 222.2 nmol/L for 25(OH)D_3_ and 13.7% at 18.1 nmol/L, 7.5% at 42.7 nmol/L and 6.4% at 127.3 nmol/L for 25(OH)D_2_. The mean bias of dried blood spot vs. serum 25(OH)D_3_ concentrations over the period 2018 to 2021 was 4.0% and the limits of quantitation were 7.5 nmol/L for 25(OH)D_3_ and 2.8 nmol/L for 25(OH)D_2_.

#### 2.6.2. Anti-S Serology Testing

Anti-S antibody titres were determined by the Clinical Immunology Service at the Institute of Immunology and Immunotherapy of the University of Birmingham (Birmingham, UK) using an ELISA that measures combined IgGAM responses to the SARS-CoV-2 trimeric S glycoprotein (product code MK654; The Binding Site [TBS], Birmingham, UK), as previously described [17]. This assay has been CE-marked with 98.3% (95% confidence interval [CI] 96.4–99.4) specificity and 98.6% (92.6–100.0) sensitivity for RT-PCR-confirmed mild-to-moderate COVID-19 [28], and has been validated as a correlate of protection against breakthrough SARS-CoV-2 infection in two populations [29,30]. A cut-off ratio relative to the TBS cut-off calibrators was determined by plotting 624 pre-2019 negatives in a frequency histogram. A cut-off coefficient was then established for IgGAM (1.31), with ratio values classed as positive (≥1) or negative (<1). Dried blood spot eluates were pre-diluted 1:40 with 0.05% PBS-Tween using a Dynex Revelation automated absorbance microplate reader (Dynex Technologies; Chantilly, VA, USA). Plates were developed after 10 min using 3,3′,5,5′-tetramethylbenzidine core, and orthophosphoric acid used as a stop solution (both TBS). Optical densities at 450 nm were measured using the Dynex Revelation.

#### 2.6.3. SARS-CoV-2 Neutralising Antibody

Serum titres of neutralising antibodies to SARS-CoV-2 were measured as previously described using an authentic virus (Wuhan Hu-1 strain) neutralisation assay [31].

#### 2.6.4. Whole Blood Stimulation Assay

Peripheral blood was collected into heparinised tubes and cultured in the presence or absence of *E. coli* lipopolysaccharide (LPS, 1–1000 ng/mL; Invivogen) or PepTivator^®^ SARS-CoV-2 Prot_S Complete (Miltenyi Biotec; 1 µg/mL) for 24 h at 37 °C in 5% CO_2_. Supernatants were harvested and stored at −80 °C pending determination of cytokine concentrations by cytometric bead array.

#### 2.6.5. Cytometric Bead Array

The cytometric bead array of whole blood assay supernatants was carried out according to the manufacturer’s protocol (BD Biosciences). The cytokines assessed were CXCL8 (IL-8), IL-6, IFN-γ and TNF. Samples were analysed using the ACEA Novocyte 3000 flow cytometer (Agilent). The lower limit of detection was 1.5 pg/mL.

#### 2.6.6. Peripheral Blood Mononuclear Cell (PBMC) Isolation

PBMCs were isolated from heparinised blood using Ficoll (Merck Life Science, Darmstadt, Germany) density gradient, washed twice in Hanks Balanced Salt Solution (Merck Life Science) and cryopreserved in 10% DMSO in Fetal Calf Serum (Invitrogen, Waltham, MA, USA).

#### 2.6.7. PBMC Stimulation Assay

Cryopreserved PBMCs were recovered and stimulated with nothing (negative control) or PepTivator^®^ SARS-CoV-2 Prot_S Complete (Miltenyi Biotec, Bergisch Gladbach, Germany; 1 µg/mL) or 1 µg/mL of soluble CD3 Monoclonal Antibody (OKT3), Functional Grade (Invitrogen) for 1 h at 37 °C in a 5% CO_2_. Brefeldin A (2.5 µg/mL) was then added to the cells, which were incubated for a further 15 h at 37 °C in 5% CO_2_.

#### 2.6.8. Flow Cytometric Analysis

Stimulated cells were collected, and the cell surface stained for CD3 (HIT3a), CD4 (RPA-T4), CD8 (SK1), CD27 (O232), CD45RA (HI100) and Zombie NIR^TM^ viability dye (Biolegend, San Diego, CA, USA) in the presence of Brilliant Buffer (BD Biosciences, Franklin Lakes, NJ, USA). Cells were washed and fixed in Intracellular Fixation Buffer (eBioscience); permeabilised in eBioscience Permeablization Buffer; stained for intracellular IL-2 (JES6-5H4), IFN-γ (4S.B3) and TNF (Mab11, Biolegend); washed and assessed using the ACEA Novocyte 3000 flow cytometer (Agilent). Data were analysed using FlowJo Version X (BD Biosciences).

### 2.7. Outcomes

Outcomes for sub-study 1 were the time to breakthrough SARS-CoV-2 infection, and the proportion of participants experiencing an episode of breakthrough infection during follow-up. Follow-up for this efficacy analysis began 14 days after participants received a second vaccine dose and participants were censored either at the time of breakthrough infection, 13 days after their booster dose or at 6 months’ follow-up, whichever occurred earlier. Outcomes for sub-study 2 were anti-S titres and the proportion of participants with detectable anti-S antibodies after vaccination. Outcomes for sub-study 3 were neutralising antibody titres; concentrations of IFN-γ, TNF, IL-6 and CXCL8 in supernatants of S peptide- and LPS-stimulated whole blood; percentages of S-peptide and CD3-stimulated T cell subsets staining positive for IFN-γ, IL-2 and TNF; and percentages of T cell subsets with naïve, central memory, effector memory and terminally differentiated effector memory cells re-expressing CD45RA (EMRA) phenotypes after vaccination.

### 2.8. Statistical Methods

The trial sample size was calculated using https://mjgrayling.shinyapps.io/multiarm/ (accessed on 17 August 2022) [32], and predicated on the numbers needed to detect a 20% reduction in the proportion of participants experiencing one or more acute respiratory infections with 84% marginal power and 5% type 1 error rate, as described elsewhere [21].

Statistical analyses were performed using Stata version 17.0. Pairwise comparisons were made between each intervention arm separately vs. the no-offer arm, and between pooled data from participants randomised to either intervention arm vs. the no-offer arm. The time to breakthrough SARS-CoV-2 infections was compared between study arms using the Cox regression, with adjustment for factors we have previously reported to be risk factors for breakthrough SARS-CoV-2 infection: age, sex, educational attainment, frontline worker status, number of people per bedroom, sharing a home with schoolchildren (5–15 years), primary vaccination course, previous SARS-CoV-2 infection, season of first vaccination, inter-dose interval, use of anticholinergics, weekly visits to or from other households, weekly visits to indoor public places other than shops and local weekly SARS-CoV-2 incidence (according to participants’ area of residence). Treatment effects were presented as adjusted hazard ratios (aHRs) with 95% CIs. Proportions of participants experiencing breakthrough SARS-CoV-2 infection were compared between arms using a logistic regression, with adjustment for the same covariates and presentation of treatment effects as adjusted odds ratios (aORs) with 95% CIs. A linear regression was used to estimate inter-arm geometric mean ratios (GMRs), with 95% CIs and associated pairwise *p*-values, for log-transformed antibody titres, cytokine concentrations, and percentages of T cell subsets staining positive for intracellular cytokines and exhibiting different phenotypes, with adjustment for factors we have previously shown to be determinants of post-vaccination anti-S titres [17]: age, sex, ethnicity, body mass index, days from second vaccine dose to DBS sample, pre-vaccination serostatus, general health, inter-dose interval and primary vaccination course. For ease of interpretation, estimated GMRs were expressed as adjusted percentage differences. For immunological outcomes, correction for multiple comparisons was performed on families of pairwise *p*-values using the Benjamini and Hochberg method with a false discovery rate of 5% [33].

We conducted a sensitivity analysis for outcomes of breakthrough SARS-CoV-2 infection, anti-S titre and S peptide-stimulated IFN-γ, excluding data from participants who reported a SARS-CoV-2 infection prior to vaccination. We also conducted an exploratory responder analysis, restricted to participants randomised to either intervention arm for whom end-trial 25(OH)D levels were available, comparing end-trial anti-S IgGAM titres, neutralizing antibody titres and S peptide-stimulated IFN-γ concentrations between those who did vs. did not attain end-trial 25(OH)D concentrations >75 nmol/L.

## 3. Results

### 3.1. Participants

Of 6200 CORONAVIT trial participants, 2808 (45.3%) received a primary course of SARS-CoV-2 vaccination and contributed data to sub-study 1 (vaccine efficacy analysis); of these, 1945 (69.3%) received two doses of ChAdOx1 nCoV-19 and 863 (30.7%) received two doses of BNT162b2. First doses were administered between 16 January and 16 June 2021; the median time from the date of initiation of vitamin D supplementation to the date of the first vaccine dose was 65 days (IQR 51–89 days). In total, 1853/2808 (66.0%) participants provided a post-vaccination dried blood spot sample between 22 March 2021 and 16 November 2021 and contributed data to sub-study 2 (analysis of anti-S IgGAM titres) and 101/2808 (3.6%) provided a post-vaccination venous blood sample between 24 May 2021 and 12 August 2021 and contributed data to the analysis of neutralising antibody and cellular responses to SARS-CoV-2 vaccination (sub-study 3; Figure 1).

Table 1 shows the baseline characteristics of participants included in the vaccine efficacy analysis (sub-study 1) by allocation. The median age was 61.9 years, 65.8% were female and 96.4% were of White ethnic origin. Among the participants whose baseline vitamin D status was tested, the mean 25(OH)D concentration was 39.9 nmol/L, and all had 25(OH)D concentrations below 75 nmol/L. Characteristics were balanced between the three trial arms, except for the proportions of participants with pre-vaccination SARS-CoV-2 infection (4.2 vs. 5.9 vs. 2.9% in no offer vs. 800 IU/day vs. 3200 IU/day arms, respectively).

The baseline characteristics for participants additionally contributing data to analyses of anti-S titres (sub-study 2) and neutralising antibody or cellular responses (sub-study 3) are presented in Appendix A, respectively: these were also balanced between trial arms. Among participants for whom end-study measurements of vitamin D status were available, mean follow-up 25(OH)D concentrations were significantly elevated in the lower-dose vs. no-offer group (82.5 (standard deviation 18.9) vs. 53.6 (25.2) nmol/L; mean difference 28.8 nmol/L, 95% CI 22.8–34.8) and in the higher-dose vs. no offer group (105.4 (23.5) vs. 53.6 (25.2) nmol/L; mean difference 51.7 nmol/L, 45.1–58.4; Figure 2A).

### 3.2. Breakthrough SARS-CoV-2 Infection

Breakthrough SARS-CoV-2 infection occurred in 174 sub-study 1 participants, with no significant inter-arm difference in time to event (lower-dose vs. no offer: median 160 (IQR 114–192) vs. 161 (131–193) days to infection, aHR 1.28, 95% CI 0.89–1.84, *p* = 0.19; higher-dose vs. no offer: 146 (88–189) vs. 160 (131–193) days to infection, aHR 1.17, 0.81–1.70, *p* = 0.40; Figure 3A,B). The results were similar when pooling data from both intervention arms (any offer vs. no offer: 153 (106–192) vs. 161 (131–193) days to infection, aHR 1.24, 0.89–1.71, *p* = 0.20; Figure 3C). Consistent with these findings, the proportions of participants experiencing a breakthrough infection did not differ by allocation (Appendix A). Five cases of COVID-19 (three in the 3200 vitamin D group and two controls) precipitated hospitalization, and none were fatal.

### 3.3. Immunological Outcomes

No inter-arm differences in the mean post-vaccination titres of combined anti-S IgGAM antibodies were seen, either when each intervention arm was compared separately to the no-offer arm (Table 2, Figure 2B) or when pooled data from both intervention arms were compared to the no-offer arm (Appendix A). Neither was there any inter-arm difference in the proportions of participants with detectable post-vaccination anti-S IgGAM antibodies (Appendix A). In the subset of participants who provided a venous blood sample for analysis, we found no significant inter-arm differences in mean neutralising antibody titres or in any antigen-specific cellular response investigated after correction for multiple testing, either when lower-dose or higher-dose arms were compared separately to the no-offer arm (Table 2, Figure 2C,D) or when pooled data from both intervention arms were compared to the no-offer arm (Appendix A).

### 3.4. Sensitivity Analysis

Excluding participants with previous SARS-CoV-2 infection did not substantially affect our findings on breakthrough SARS-CoV-2 infection (Appendix A), anti-S IgGAM titres (Appendix A) or S peptide-stimulated IFN-γ (Appendix A).

### 3.5. Exploratory Responder Analysis

An exploratory responder analysis, restricted to participants randomised to either intervention arm for whom end-trial 25(OH)D levels were available, showed a trend towards higher end-trial anti-S IgGAM titres in participants who did vs. did not attain end-trial 25(OH)D concentrations ≥75 nmol/L (adjusted percentage difference 17.6%, 95% CI -0.3% to 38.8%). No differences in neutralizing antibody titres or S peptide-stimulated IFN-γ concentrations were seen between those with higher vs. lower end-trial 25(OH)D concentrations however (Appendix A).

## 4. Discussion

We reported the findings of sub-studies nested within a randomised controlled trial to investigate the effects of vitamin D supplementation on SARS-CoV-2 vaccine efficacy and immunogenicity. All participants had 25(OH)D concentrations below 75 nmol/L at baseline, and supplementation with both 800 IU and 3200 IU of vitamin D per day was effective in elevating end-study 25(OH)D concentrations in the intervention groups. However, improvements in vitamin D status were not associated with inter-arm differences in the risk of breakthrough SARS-CoV-2 infection, post-vaccination titres of anti-S or neutralising antibodies or any cellular immune response investigated.

Null results from the current intervention study are in keeping with those from two observational studies in the field [19,20], but contrast with findings from two others that report positive results [17,18]. Of these, one reported an association between higher post-vaccination anti-S titres and circulating 25(OH)D concentrations of more than 50 nmol/L in a cohort of health care workers [18]. The other, a population-based study conducted in UK adults, found an independent association between vitamin D supplement use and the reduced risk of anti-S seronegativity following SARS-CoV-2 vaccination [17]. These positive associations may have arisen as a result of unmeasured or residual confounding, or type 1 error. The fact that no inter-arm difference in anti-S titres was seen in the current study supports the interpretation that the null result from the current analysis is valid, since it is biologically implausible that vitamin D would affect the proportion of seronegative participants but not the mean anti-S titre. The null findings presented here also contrast with the results of our previous intervention study [14], in which we showed that vitamin D replacement in older adults with baseline 25(OH)D levels below 75 nmol/L boosted antigen-specific immunity and reduced inflammatory responses to a cutaneous VZV antigen challenge. Divergent findings between these two intervention studies may reflect differences in the compartment studied (peripheral blood vs. skin), immunological stimulus (SARS-CoV-2 vaccination vs. VZV antigen challenge) or the regimen of vitamin D administered (800 or 3200 IU/day for at least 1 month vs. 6400 IU/day for 14 weeks before stimulation).

Our study has several strengths. Participants had a sub-optimal vitamin D status at baseline, and interventions were effective in elevating 25(OH)D levels into the physiological range. The large sample sizes of sub-studies 1 and 2, together with the substantial number of breakthrough SARS-CoV-2 infections arising in sub-study 1, provided good power to detect the effects of the intervention. We also investigated a combination of clinical and immunological outcomes, with detailed characterisation of both humoral and cellular responses: the fact that our results were consistent across a broad range of outcomes strengthens the interpretation that our null results are valid.

Our study also has limitations. Randomisation could not be stratified according to sub-study participation, since eligibility for inclusion in one or more sub-studies was contingent on factors arising during follow-up. However, baseline characteristics were similar between arms for all sub-studies, and we adjusted for multiple factors influencing vaccine efficacy and immunogenicity, thereby minimising the potential for confounding. There was a baseline imbalance in pre-vaccination SARS-CoV-2 status between study arms. We accounted for this in the analysis by adjustment in the main model, and by conducting a sensitivity analysis excluding data from participants who had a pre-vaccination SARS-CoV-2 infection at baseline: these complementary approaches yielded similar results. Our study was open label, and participants were therefore aware of their allocation; however, laboratory staff were blinded to participant allocation, thereby removing the potential for observer bias to influence the assessment of immunological outcomes. No restrictions regarding vitamin D intake were stipulated for participants randomised to the no-offer arm; however, participants in this group who reported taking supplements were excluded from the analyses presented here. The fact that the attained 25(OH)D concentrations differed markedly between arms suggests that we were successful in excluding participants in the no-offer arm who used off-trial vitamin D supplements during follow-up. Just five cases of COVID-19 arising in sub-study 1 participants precipitated hospitalization, and none were fatal: thus, the trial lacked the power to detect the effects of vitamin D on the efficacy of SARS-CoV-2 vaccination to prevent severe disease specifically. Finally, the lack of a measurement of baseline vitamin D status among participants in the no-offer arm precludes sub-group analyses to test for this as an effect modifier. Although all participants tested had baseline 25(OH)D concentrations below 75 nmol/L, we cannot rule out an effect in the sub-groups of participants with the lowest baseline 25(OH)D concentrations.

In conclusion, we report that the daily administration of 800 IU or 3200 IU vitamin D3 was effective in elevating circulating 25(OH)D concentrations, but that neither dose influenced SARS-CoV-2 vaccine efficacy or immunogenicity. Our findings do not support the use of vitamin D supplements as an adjunct to SARS-CoV-2 vaccination.

## Figures and Tables

**Figure 1 nutrients-14-03821-f001:**
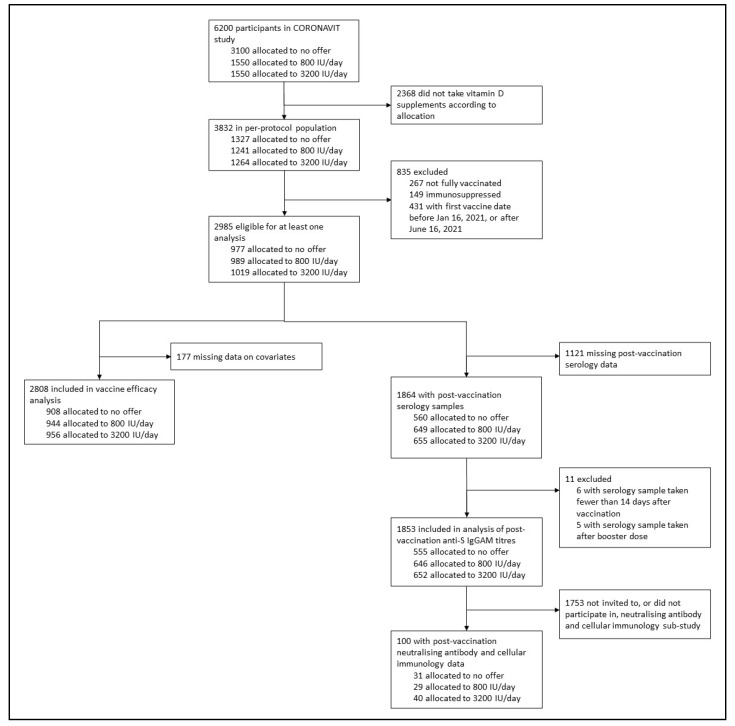
Participant flow.

**Figure 2 nutrients-14-03821-f002:**
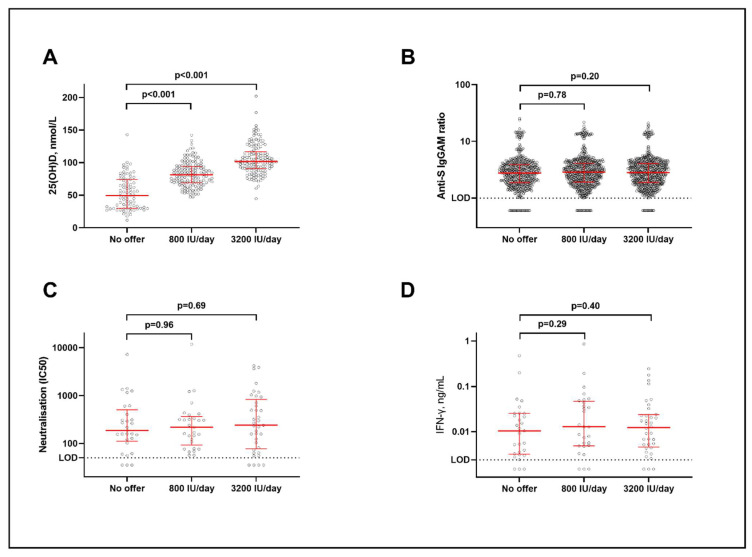
Biochemical and immunological outcomes by allocation. (**A**) End-study 25(OH)D concentrations. (**B**) Anti-S IgGAM titres. (**C**) Neutralising antibody titres. (**D**) IFN-γ concentrations in supernatants from S peptide-stimulated whole blood. Horizontal bars represent medians and interquartile ranges. *p* values from unpaired *t* tests (**A**) and multiple linear regression with adjustment for covariates as described in Methods (**B**–**D**). Here, 25(OH)D = 25-hydroxyvitamin D. Anti-S IgGAM = combined anti-Spike IgG, IgA and IgM response. IFN = interferon. LOD = limit of detection.

**Figure 3 nutrients-14-03821-f003:**
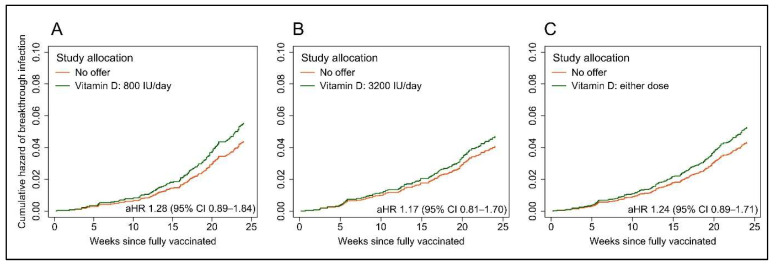
Cumulative hazard plots showing risk of breakthrough SARS-CoV-2 infection by allocation. (**A**), Offer of 800 IU vitamin D3/day vs. no offer. (**B**), Offer of 3200 IU vitamin D3/day vs. no offer. (**C**), Either offer (pooled) vs. no offer. Hazard ratios adjusted for age, sex, educational attainment, frontline worker status, number of people per bedroom, schoolchildren (5–15 years) at home with participant, primary vaccination course, previous SARS-CoV-2 infection, season of first vaccination, inter-dose interval, use of anticholinergics, weekly visits to or from other households, weekly visits to indoor public places other than shops and local weekly SARS-CoV-2 incidence in participants’ area of residence. aHR, adjusted hazard ratio.

**Table 1 nutrients-14-03821-t001:** Baseline characteristics of participants contributing data to vaccine efficacy analysis (sub-study 1), by allocation.

	Overall (*n* = 2808)	No Offer (*n* = 908)	800 IU/Day Offer (*n* = 944)	3200 IU/Day Offer (*n* = 956)
Age, years				
Median (IQR)	61.9 (54.1–68.5)	61.8 (53.6–68.1)	61.4 (54.2–68.3)	62.5 (54.1–69.1)
Range	16.6–87.2	18.4–85.9	20.9–83.1	16.6–87.2
Sex				
Female	1848 (65.8%)	579 (63.8%)	634 (67.2%)	635 (66.4%)
Male	960 (34.2%)	329 (36.2%)	310 (32.8%)	321 (33.6%)
Ethnicity				
White	2706 (96.4%)	885 (97.5%)	910 (96.4%)	911 (95.3%)
Asian/Asian British	25 (0.9%)	9 (1.0%)	5 (0.5%)	11 (1.2%)
Black/African/Caribbean/Black British	14 (0.5%)	2 (0.2%)	6 (0.6%)	6 (0.6%)
Mixed/Multiple/Other	63 (2.2%)	12 (1.3%)	23 (2.4%)	28 (2.9%)
Body mass index, kg/m²				
<25	1304 (46.5%)	404 (44.5%)	440 (46.8%)	460 (48.1%)
25–30	949 (33.8%)	314 (34.6%)	307 (32.6%)	328 (34.3%)
>30	551 (19.7%)	189 (20.8%)	194 (20.6%)	168 (17.6%)
Self-assessed general health				
Excellent	617 (22.0%)	203 (22.4%)	196 (20.8%)	218 (22.8%)
Very good	1186 (42.3%)	386 (42.5%)	404 (42.8%)	396 (41.5%)
Good	721 (25.7%)	213 (23.5%)	257 (27.2%)	251 (26.3%)
Fair	222 (7.9%)	86 (9.5%)	67 (7.1%)	69 (7.2%)
Poor	61 (2.2%)	20 (2.2%)	20 (2.1%)	21 (2.2%)
Medically diagnosed disease				
Hypertension	577 (20.5%)	191 (21.0%)	190 (20.1%)	196 (20.5%)
Diabetes	135 (4.8%)	50 (5.5%)	35 (3.7%)	50 (5.2%)
Heart disease	113 (4.0%)	41 (4.5%)	39 (4.1%)	33 (3.5%)
Asthma	394 (14.0%)	120 (13.2%)	160 (16.9%)	114 (11.9%)
COPD	45 (1.6%)	18 (2.0%)	15 (1.6%)	12 (1.3%)
Pre-vaccination SARS-CoV-2 infection ^(1)^	121 (4.3%)	38 (4.2%)	55 (5.8%)	28 (2.9%)
Type of vaccine administered, primary course ^(2)^				
2 × ChAdOx1	1945 (69.3%)	632 (69.6%)	673 (71.3%)	640 (66.9%)
2 × BNT162b2	863 (30.7%)	276 (30.4%)	271 (28.7%)	316 (33.1%)
Inter-dose interval, days	77 (69–79)	77 (68–79)	77 (69–79)	77 (69–79)
Mean 25(OH)D, nmol/L (SD) [range] ^(3)^	39.9 (14.5) [10.3–74.9]	-- ^(4)^	39.6 (14.7) [10.3–74.8]	40.2 (14.4) [10.3–74.9]
<25.0	309 (11.0%)	-- ^(4)^	164 (17.4%)	145 (15.2%)
25.0 to <50.0	1104 (39.3%)	-- ^(4)^	543 (57.5%)	561 (58.7%)
50.0 to <75.0	482 (17.2%)	-- ^(4)^	235 (24.9%)	247 (25.8%)
≥75.0	0	-- ^(4)^	0	0
Not determined	913 (32.5%)	908 (100.0%)	2 (0.2%)	3 (0.3%)

Data are *n* (%) or median (IQR) unless specified otherwise. Abbreviations: IQR, interquartile range; SD, standard deviation; 25(OH)D, 25-hydroxyvitamin D. ^(1)^ Reported swab test-positive infection by RT-PCR or antigen testing. ^(2)^ Vaccine efficacy analysis restricted to participants receiving a full course of either ChAdOx1 or BNT162b2. ^(3)^ Missing values: 25(OH)D concentration missing for three participants in 3200 IU/day arm and two participants in 800 IU/day arm. ^(4)^ Baseline 25(OH)D not determined for participants randomly assigned to the no-offer arm.

**Table 2 nutrients-14-03821-t002:** Immunological outcomes by allocation.

	No Offer	800 IU/Day Offer	3200 IU/Day Offer	800 IU/Day vs. No Offer		3200 IU/Day vs. No Offer	
				Adjusted % difference (95% CI) *	P†	Adjusted % difference (95% CI) *	P†
Sub-study 2							
Anti-S IgGAM ratio	2.8 (1.9 to 3.9) [*n* = 555]	2.9 (1.9 to 4.1) [*n* = 646]	2.8 (1.9 to 4.1) [*n* = 652]	−20.6% (−84.9 to 316.6)	0.781	−64.4% (−92.9 to 78.5)	0.204
Sub-study 3							
Neutralising antibody titre	186.9 (119.8 to 406.7) [*n* = 29]	219.0 (107.0 to 333.0) [*n* = 29]	241.2 (81.2 to 713.8) [*n* = 37]	1.2% (−40.0 to 70.7)	0.963	−10.8% (−49.5 to 57.5)	0.686
S peptide-stimulated IFN-γ in whole blood supernatant, ng/mL	0.011 (0.003 to 0.025) [*n* = 29]	0.013 (0.005 to 0.046) [*n* = 28]	0.013 (0.005 to 0.024) [*n* = 37]	54.7% (−31.3 to 248.2)	0.285	49.2% (−41.8 to 282.2)	0.396
S peptide-stimulated TNF in whole blood supernatant, ng/mL	0.000 (0.000 to 0.003) [*n* = 29]	0.000 (0.000 to 0.000) [*n* = 28]	0.000 (0.000 to 0.027) [*n* = 39]	−18.9% (−72.5 to 139.0)	0.700	−68.4% (−90.0 to −0.5)	0.049
S peptide-stimulated IL-6 in whole blood supernatant, ng/mL	0.136 (0.012 to 0.719) [*n* = 28]	0.127 (0.000 to 1.246) [*n* = 27]	0.080 (0.000 to 4.273) [*n* = 39]	−45.3% (−89.7 to 189.8)	0.471	−60.3% (−94.7 to 197.1)	0.360
S peptide-stimulated CXCL8 in whole blood supernatant, ng/mL	0.790 (0.254 to 2.563) [*n* = 26]	0.893 (0.295 to 2.577) [*n* = 27]	1.416 (0.388 to 12.016) [*n* = 39]	70.0% (−50.6 to 485.4)	0.393	−12.7% (−80.1 to 283.7)	0.854
LPS-stimulated IFN-γ in whole blood supernatant, ng/mL	0.020 (0.004 to 0.067) [*n* = 29]	0.027 (0.009 to 0.062) [*n* = 28]	0.028 (0.006 to 0.109) [*n* = 37]	44.9% (−52.7 to 343.4)	0.509	−14.9% (−67.8 to 125.0)	0.739
LPS-stimulated TNF in whole blood supernatant, ng/mL	0.542 (0.465 to 0.919) [*n* = 29]	0.597 (0.350 to 0.923) [*n* = 28]	0.473 (0.302 to 0.836) [*n* = 40]	−57.8% (−85.1 to 19.2)	0.101	−17.1% (−51.5 to 41.7)	0.484
LPS-stimulated IL-6 in whole blood supernatant, ng/mL	54.768 (32.362 to 73.147) [*n* = 29]	43.202 (28.838 to 79.264) [*n* = 28]	46.142 (32.146 to 91.936) [*n* = 40]	10.4% (−26.3 to 65.5)	0.625	−15.3% (−45.4 to 31.3)	0.449
LPS-stimulated CXCL8 in whole blood supernatant, ng/mL	4.231 (2.322 to 6.373) [*n* = 28]	2.642 (1.540 to 4.213) [*n* = 28]	2.732 (2.169 to 4.392) [*n* = 40]	−9.4% (−39.7 to 36.3)	0.631	−29.9% (−57.6 to 16.0)	0.162
Percentage of S peptide-stimulated CD3 + CD4+ cells positive for intracellular IFN-γ	0.00% (0.00 to 0.02) [*n* = 29]	0.00% (0.00 to 0.02) [*n* = 27]	0.00% (0.00 to 0.02) [*n* = 37]	−31.7% (−59.6 to 15.4)	0.151	−17.5% (−60.1 to 70.9)	0.598
Percentage of CD3-stimulated CD3 + CD4+ cells positive for intracellular IFN-γ	0.04% (0.00 to 0.12) [*n* = 29]	0.09% (0.01 to 0.23) [*n* = 26]	0.12% (0.04 to 0.42) [*n* = 37]	285.6% (26.6 to 1074.1)	0.019	156.1% (−30.8 to 847.2)	0.154
Percentage of S peptide-stimulated CD3 + CD8+ cells positive for intracellular IFN-γ	0.00% (0.00 to 0.04) [*n* = 29]	0.00% (0.00 to 0.08) [*n* = 27]	0.01% (0.00 to 0.06) [*n* = 37]	10.8% (−48.4 to 137.8)	0.788	50.2% (−41.5 to 285.6)	0.389
Percentage of CD3-stimulated CD3 + CD8+ cells positive for intracellular IFN-γ	0.62% (0.18 to 2.15) [*n* = 29]	0.93% (0.40 to 2.09) [*n* = 26]	0.95% (0.25 to 1.96) [*n* = 37]	70.3% (−35.4 to 349.0)	0.276	106.1% (−29.5 to 501.9)	0.181
Percentage of S peptide-stimulated CD3 + CD4+ cells positive for intracellular IL-2	0.00% (0.00 to 0.02) [*n* = 29]	0.01% (0.00 to 0.03) [*n* = 27]	0.01% (0.00 to 0.02) [*n* = 37]	23.7% (−18.6 to 88.0)	0.312	36.4% (−12.9 to 113.7)	0.170
Percentage of CD3-stimulated CD3 + CD4+ cells positive for intracellular IL-2	0.12% (0.09 to 0.25) [*n* = 29]	0.17% (0.06 to 0.32) [*n* = 26]	0.18% (0.09 to 0.35) [*n* = 37]	45.7% (−18.8 to 161.7)	0.202	47.8% (−36.2 to 242.8)	0.353
Percentage of S peptide-stimulated CD3 + CD8+ cells positive for intracellular IL-2	0.00% (0.00 to 0.00) [*n* = 29]	0.00% (0.00 to 0.00) [*n* = 27]	0.00% (0.00 to 0.00) [*n* = 37]	43.1% (−0.6 to 106.2)	0.054	28.6% (−21.1 to 109.6)	0.304
Percentage of CD3-stimulated CD3 + CD8+ cells positive for intracellular IL-2	0.13% (0.05 to 0.26) [*n* = 29]	0.20% (0.11 to 0.47) [*n* = 26]	0.17% (0.08 to 0.31) [*n* = 37]	56.0% (−19.6 to 202.7)	0.184	78.9% (−23.3 to 317.4)	0.173
Percentage of S peptide-stimulated CD3 + CD4+ cells positive for intracellular TNF	0.00% (0.00 to 0.00) [*n* = 29]	0.00% (0.00 to 0.04) [*n* = 27]	0.00% (0.00 to 0.02) [*n* = 37]	22.6% (−36.0 to 135.2)	0.532	63.0% (−28.8 to 273.1)	0.241
Percentage of CD3-stimulated CD3 + CD4+ cells positive for intracellular TNF	0.37% (0.19 to 0.67) [*n* = 29]	0.67% (0.37 to 1.08) [*n* = 26]	0.74% (0.22 to 1.37) [*n* = 37]	80.4% (−15.2 to 283.9)	0.123	101.9% (−24.4 to 439.2)	0.156
Percentage of S peptide-stimulated CD3 + CD8+ cells positive for intracellular TNF	0.02% (0.00 to 0.06) [*n* = 29]	0.03% (0.00 to 0.11) [*n* = 27]	0.00% (0.00 to 0.06) [*n* = 37]	−41.8% (−76.6 to 45.0)	0.239	−6.5% (−70.6 to 197.8)	0.908
Percentage of CD3-stimulated CD3 + CD8+ cells positive for intracellular TNF	1.45% (0.72 to 3.84) [*n* = 29]	2.57% (1.56 to 4.26) [*n* = 26]	2.38% (0.82 to 4.40) [*n* = 37]	129.4% (−11.5 to 495.1)	0.086	176.1% (−10.3 to 749.4)	0.075
Percentage of CD3 + CD4+ cells with naive phenotype	35.8% (21.7 to 48.7) [*n* = 29]	31.9% (20.2 to 42.6) [*n* = 27]	36.8% (23.6 to 43.1) [*n* = 37]	−0.2% (−23.5 to 30.2)	0.988	−8.6% (−32.0 to 22.8)	0.541
Percentage of CD3 + CD8+ cells with naive phenotype	32.5% (22.9 to 42.6) [*n* = 29]	24.4% (19.5 to 32.7) [*n* = 27]	24.8% (19.6 to 36.3) [*n* = 37]	−14.0% (−32.1 to 8.9)	0.204	−14.5% (−34.4 to 11.5)	0.241
Percentage of CD3 + CD4+ cells with central memory phenotype	33.5% (26.6 to 40.6) [*n* = 29]	39.6% (31.2 to 46.5) [*n* = 27]	35.6% (30.7 to 44.9) [*n* = 37]	14.4% (−6.8 to 40.4)	0.194	24.3% (3.4 to 49.3)	0.021
Percentage of CD3 + CD8+ cells with central memory phenotype	20.5% (13.9 to 28.8) [*n* = 29]	24.7% (15.9 to 33.2) [*n* = 27]	22.6% (11.6 to 28.1) [*n* = 37]	10.1% (−20.8 to 53.0)	0.559	35.5% (0.0 to 83.7)	0.050
Percentage of CD3 + CD4+ cells with effector memory phenotype	11.7% (7.3 to 15.7) [*n* = 29]	14.4% (9.8 to 17.3) [*n* = 27]	13.0% (8.8 to 17.9) [*n* = 37]	8.4% (−17.7 to 42.9)	0.558	16.6% (−13.6 to 57.4)	0.306
Percentage of CD3 + CD8+ cells with effector memory phenotype	8.2% (5.6 to 12.7) [*n* = 29]	12.2% (9.5 to 15.7) [*n* = 27]	11.4% (6.5 to 15.1) [*n* = 37]	32.3% (−3.7 to 81.7)	0.083	44.4% (3.3 to 102.0)	0.033
Percentage of CD3 + CD4+ cells with EMRA phenotype	3.9% (2.4 to 7.9) [*n* = 29]	3.6% (2.4 to 6.1) [*n* = 27]	3.1% (1.7 to 5.2) [*n* = 37]	−35.6% (−59.0 to 1.2)	0.056	−20.5% (−53.2 to 35.1)	0.388
Percentage of CD3 + CD8+ cells with EMRA phenotype	14.9% (10.3 to 29.7) [*n* = 29]	18.5% (9.5 to 25.9) [*n* = 27]	20.7% (14.0 to 28.7) [*n* = 37]	15.6% (−16.0 to 59.2)	0.366	0.2% (−33.9 to 52.0)	0.992

Data are median (IQR) [*n*] unless otherwise specified. Values below the limit of detection are presented as 0. * Adjusted for age, sex, ethnicity, body mass index, pre-vaccination anti-S IgGAM, days from second vaccine dose to DBS sample, general health, inter-dose interval and primary vaccination course. † Correction for multiple testing using the Benjamin and Hochberg procedure provides a critical *p* value of 0.0017. EMRA, terminally differentiated effector memory cells re-expressing CD45RA.

## Data Availability

Anonymised participant-level data will be made available on reasonable request to a.martineau@qmul.ac.uk, subject to the terms of Research Ethics Committee and Sponsor approval.

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
