# Peer review of "Vitamin D Supplementation Does Not Influence SARS-CoV-2 Vaccine Efficacy or Immunogenicity: Sub-Studies Nested within the CORONAVIT Randomised Controlled Trial"

_nutrients, 2022, doi:10.3390/nu14183821_

Round 1

Reviewer 1 Report

The authors nested three sub-studies conducted within the CORONAVIT randomized controlled trial. Sub-study 1 performed vaccine efficacy analysis to assess the influence of vitamin D supplementation on the risk of breakthrough COVID-19 infection among immunocompetent participants vaccinated with two SARS-CoV-2 vaccine doses. Sub-study 2 performed a dried blood spot analysis that assessed the impact of supplementing vitamin D on the combined immunoglobulin G (IgG), IgA, and IgM (IgGAM) antibody responses to the SARS-CoV-2 spike (S) protein.  In addition, sub-study 3 investigated the effects of vitamin D supplementation on neutralising antibodies and cell responses.

The study design was concise and could effectively address the question of whether vitamin D supplements can contribute to the effectiveness of the SARS-CoV-2 vaccine in clinical scenarios.

The study population was limited to immune-competent individuals and participants residing in the United Kingdom. Vitamin D food fortification is common in European countries, but many other countries in the world haven't performed Vitamin D food fortification. I suggest the authors need to emphasize this issue in Discussion. The cutoff of 75nmol/L may be too high to see any benefit of Vitamin D supplementation, especially for those countries where vitamin D deficiency is prevalent. Besides evaluating the efficacy of the vaccine in the prevention of breakthrough infection, the authors may also consider assessing the vaccine's effectiveness in preventing an exposed person from developing serious diseases that require hospitalization and lead to mortality.

Author Response

Reviewer #1: The authors nested three sub-studies conducted within the CORONAVIT randomized controlled trial. Sub-study 1 performed vaccine efficacy analysis to assess the influence of vitamin D supplementation on the risk of breakthrough COVID-19 infection among immunocompetent participants vaccinated with two SARS-CoV-2 vaccine doses. Sub-study 2 performed a dried blood spot analysis that assessed the impact of supplementing vitamin D on the combined immunoglobulin G (IgG), IgA, and IgM (IgGAM) antibody responses to the SARS-CoV-2 spike (S) protein.  In addition, sub-study 3 investigated the effects of vitamin D supplementation on neutralising antibodies and cell responses. The study design was concise and could effectively address the question of whether vitamin D supplements can contribute to the effectiveness of the SARS-CoV-2 vaccine in clinical scenarios.

Reviewer 1, Comment 1: The study population was limited to immune-competent individuals and participants residing in the United Kingdom. Vitamin D food fortification is common in European countries, but many other countries in the world haven't performed Vitamin D food fortification. I suggest the authors need to emphasize this issue in Discussion.

RESPONSE: Unlike in many other European countries, vitamin D fortification of foods is not mandatory in the UK. Thus, we have not edited the Discussion in response to this point.

Reviewer 1, Comment 2: The cutoff of 75nmol/L may be too high to see any benefit of Vitamin D supplementation, especially for those countries where vitamin D deficiency is prevalent. 

RESPONSE: We acknowledge that benefits of vitamin D supplementation may be greater in or restricted to those with baseline 25D levels <50 or <25 nmol/L. However, we are unable to test this hypothesis in the current trial because baseline 25(OH)D concentrations were not measured in participants randomised to the control arm of the trial. The following text in the ‘Limitations’ section of the Discussion addresses this point:

“Finally, the lack of a measurement of baseline vitamin D status among participants in the no-offer arm precludes sub-group analyses to test for this as an effect modifier. Although all participants tested had baseline 25(OH)D concentrations below 75 nmol/L, we cannot rule out an effect in sub-groups of participants with the lowest baseline 25(OH)D concentrations.”

Reviewer 1, Comment 3: Besides evaluating the efficacy of the vaccine in the prevention of breakthrough infection, the authors may also consider assessing the vaccine's effectiveness in preventing an exposed person from developing serious diseases that require hospitalization and lead to mortality.

RESPONSE: The reviewer’s point is well made. However, just 5 cases of COVID-19 (three in the 3200 vitamin D group and two controls) precipitated hospitalization, and none were fatal. Thus we are not powered to investigate the effect of vitamin D on vaccine-induced protection against disease requiring hospitalization. These details have been added to Results, and the following text has been added to the ‘limitations’ section of the Discussion:

“Just five cases of COVID-19 arising in sub-study 1 participants precipitated hospitalization, and none were fatal: thus, the trial lacked power to detect effects of vitamin D on the efficacy of SARS-CoV-2 vaccination to prevent severe disease specifically.”

Reviewer 2 Report

The authors have investigated the influence of Vitamin D supplementation on SARS-CoV-2 vaccine efficacy and immunogenicity. The study design and manuscript preparation are commendable. The study is essential to improve current medical knowledge.

There are a few minor comments:

The title may be altered so that it clearly indicates the findings of the study. The current title seems to be misleading.

Table 2 may be given as graphs.

Author Response

Reviewer 2. The authors have investigated the influence of Vitamin D supplementation on SARS-CoV-2 vaccine efficacy and immunogenicity. The study design and manuscript preparation are commendable. The study is essential to improve current medical knowledge. There are a few minor comments:

Reviewer 2, Comment 1: The title may be altered so that it clearly indicates the findings of the study. The current title seems to be misleading.

RESPONSE: We take the reviewer’s point and have edited the title so that it reads as follows:

‘Influence of Vitamin D Supplementation does not influence on SARS-CoV-2 Vaccine Efficacy and or Immunogenicity: Sub-studies Nested within the CORONAVIT Randomised Controlled Trial.’

Reviewer 2, Comment 2: Table 2 may be given as graphs.

RESPONSE: Table 2 presents data for 29 different outcome measures, with all results null. Graphical presentation of these data would require a figure with a very large number of panels, each of which would need to be very small to fit on the page – this would make interpretation more challenging. Thus, the tabular presentation seems most appropriate.

ADDITIONAL EDIT: In response to comments received on the pre-print of this manuscript (https://www.medrxiv.org/content/10.1101/2022.07.15.22277678v2 ), we have conducted an exploratory responder analysis to determine whether key outcomes differed between participants randomised to intervention who did vs. did not attain optimal 25(OH)D concentrations (i.e. 25[OH]D >75 nmol/L).

The following text has been added to Methods:

We also conducted an exploratory responder analysis, restricted to participants randomised to either intervention arm for whom end-trial 25(OH)D levels were available, comparing end-trial anti-S IgGAM titres, neutralizing antibody titres and S peptide-stimulated IFN-γ concentrations between those who did vs. did not attain end-trial 25(OH)D concentrations >75 nmol/L.

The following text has been added to Results:

3.5 Exploratory responder analysis

An exploratory responder analysis, restricted to participants randomised to either intervention arm for whom end-trial 25(OH)D levels were available, showed no statisti-cally significant differences in end-trial anti-S IgGAM titres, neutralizing antibody titres or S peptide-stimulated IFN-γ concentrations between those who did vs. did not attain end-trial 25(OH)D concentrations >75 nmol/L (Table S8, Supplementary Appendix).

And Table S8 has been added to the Supplementary Appendix.